# Restaging the Biochemical Recurrence of Prostate Cancer with [^68^Ga]Ga-PSMA-11 PET/CT: Diagnostic Performance and Impact on Patient Disease Management

**DOI:** 10.3390/cancers13071594

**Published:** 2021-03-30

**Authors:** Aloÿse Fourquet, Lucien Lahmi, Timofei Rusu, Yazid Belkacemi, Gilles Créhange, Alexandre de la Taille, Georges Fournier, Olivier Cussenot, Mathieu Gauthé

**Affiliations:** 1Department of Nuclear Medicine, Hôpital Tenon-AP-HP, Sorbonne Université, 75020 Paris, France; aloyse.fourquet@aphp.fr (A.F.); timofei.rusu@aphp.fr (T.R.); 2Department of Radiation Oncology, Hôpital Tenon-AP-HP, Sorbonne Université, 75020 Paris, France; Lucien.lahmi@gustaveroussy.fr; 3Department of Radiation Oncology and Henri Mondor Breast Center, Hôpitaux Universitaires Henri Mondor, Université Paris-Est Créteil (UEPC) et IMRB—INSERM U955 Team 21, 94000 Créteil, France; yazid.belkacemi@aphp.fr; 4Department of Radiation Oncology, Institut Curie, 75005 Paris, France; gilles.crehange@curie.fr; 5Department of Urology, Hôpitaux Universitaires Henri Mondor, Université Paris-Est Créteil (UEPC), 94000 Créteil, France; alexandre.de-la-taille@aphp.fr; 6Department of Urology, Hôpital de la Cavale Blanche, Université de Brest, 29200 Brest, France; gfournier@afu.fr; 7Department of Urology, Hôpital Tenon-AP-HP, Sorbonne Université, 75020 Paris, France; olivier.cussenot@aphp.fr; 8AP-HP Health Economics Research Unit, INSERM-UMR1153, 75004 Paris, France

**Keywords:** prostatic neoplasms, positron-emission tomography, decision making

## Abstract

**Simple Summary:**

We aimed to evaluate the diagnostic performance, impact on patient disease management, and therapy efficacy prediction of [^68^Ga]Ga-PSMA-11 PET/CT on 294 patients with biochemical recurrence of prostate cancer. We established a composite standard of truth for the imaging based on all clinical data available collected during the follow-up period with a median duration of follow-up of 17 months. Using this methodology, we found that the overall per-patient sensitivity and specificity were both 70%, the patient disease management was changed in 68% of patients, and that [^68^Ga]Ga-PSMA-11 PET/CT impacted this change in 86% of patients. The treatment carried out on the patient was considered effective in 78% of patients; in 89% of patients when guided by [^68^Ga]Ga-PSMA-11 PET/CT versus 61% of patients when not guided by [^68^Ga]Ga-PSMA-11 PET/CT.

**Abstract:**

Background: Detection rates of [^68^Ga]Ga-PSMA-11 PET/CT on the restaging of prostate cancer (PCa) patients presenting with biochemical recurrence (BCR) have been well documented, but its performance and impact on patient management have not been evaluated as extensively. Methods: Retrospective analysis of PCa patients presenting with BCR and referred for [^68^Ga]Ga-PSMA-11 PET/CT. Pathological foci were classified according to six anatomical sites and evaluated with a three-point scale according to the uptake intensity. The impact of [^68^Ga]Ga-PSMA-11 PET/CT was defined as any change in management that was triggered by [^68^Ga]Ga-PSMA-11 PET/CT. The existence of a PCa lesion was established according to a composite standard of truth based on all clinical data available collected during the follow-up period. Results: We included 294 patients. The detection rate was 69%. Per-patient sensitivity and specificity were both 70%. Patient disease management was changed in 68% of patients, and [^68^Ga]Ga-PSMA-11 PET/CT impacted this change in 86% of patients. The treatment carried out on patient was considered effective in 89% of patients when guided by [^68^Ga]Ga-PSMA-11 PET/CT versus 61% of patients when not guided by [^68^Ga]Ga-PSMA-11 PET/CT (*p* < 0.001). Conclusions: [^68^Ga]Ga-PSMA-11 PET/CT demonstrated high performance in locating PCa recurrence sites and impacted therapeutic management in nearly two out of three patients.

## 1. Introduction

Prostate cancer (PCa) is the most prevalent cancer in men worldwide, accounting for approximately 21% of all diagnosed cancers [1]. Up to 40% of patients with PCa initially treated with curative intent will experience biochemical recurrence (BCR) [2,3], which is defined following radical prostatectomy by two consecutive rising prostate-specific antigen (PSA) values >0.2 ng/mL, or after primary radiation therapy by any PSA increase ≥2 ng/mL higher than the PSA nadir value, regardless of the serum concentration of the nadir [4,5]. Accurately locating the recurrence site(s) is essential for optimizing patient management, as localized or oligometastatic recurrences could be eligible for salvage targeted treatments with curative intent, such as local therapy [6] or stereotactic radiation therapy [7]. Conventional imaging modalities, such as bone scan and computed tomography (CT), have limited utility in this setting, especially when PSA serum levels are below 10 ng/mL [8]. [^18^F]fluorocholine positron emission tomography associated with computed tomography (PET/CT) was demonstrated to have better performance than conventional imaging but may also fail to locate recurrence at low PSA levels [9].

The prostate-specific membrane antigen (PSMA) is a transmembrane protein that is over-expressed by up to 1000-fold in almost all PCa cells [10,11]. The recent introduction of PET/CT using a PSMA radioligand for imaging of PCa BCR has shown promising results due to its performance in detecting lesions, even at very low PSA levels, impacting on the therapeutic management of PCa patients [12,13,14].

Although the detection rates of ^68^Ga-PSMA PET/CT have been well documented, its sensitivity, specificity, impact on patient management, and therapy efficacy prediction have not been evaluated as extensively. 

The purpose of this study was to evaluate the diagnostic performance, impact on patient disease management, and therapy efficacy prediction of PET/CT using a PSMA ligand radiolabelled with gallium-68, the [^68^Ga]Ga-PSMA-11, on the restaging of PCa patients presenting with BCR. 

## 2. Materials and Methods

### 2.1. Population

Patients presenting with BCR of PCa who were addressed to our department for [^68^Ga]Ga-PSMA-11 PET/CT were consecutively included and retrospectively analyzed. These patients had shown no sign of distant metastases at [^18^F]fluorocholine PET/CT; for this reason, they were referred to [^68^Ga]Ga-PSMA-11 PET/CT, based on the French regulation for compassionate use of pharmaceutical, which is authorized on an individual basis by the National Medicine Agency.

Inclusion criteria for patients were as follows: 1—histologically confirmed PCa previously treated with curative intent; 2—no known history of PCa distant metastases (invaded locoregional pelvic lymph node at diagnosis was not considered as metastatic according to the 2009 TNM classification for staging PCa [15]); 3—currently presenting a biochemical recurrence defined as two consecutive rising PSA values above 0.2 ng/mL following radical prostatectomy or any PSA increase greater than or equal to 2 ng/mL higher than the PSA nadir value, regardless of the nadir value, for non-surgical first-line definitive treatments [4].

Exclusion criteria were as follows: 1—PCa with known distant metastases; 2—patients with persistent PSA after prostatectomy (PSA ≥ 0.1 ng/mL) [15] or radiation therapy (nadir PSA < 2 ng/mL with testosterone recovered if previous androgen deprivation therapy (ADT) [5]); 3—patients who were never treated with curative intent for PCa; 4—the presence of a second active neoplasm other than PCa. 

This research implied no intervention on the patient. According to French regulations, the approval of an institutional review board was not necessary for performing this retrospective analysis of already available data. Patients were informed that their data collected for the [^68^Ga]Ga-PSMA-11 PET/CT would be analyzed and published anonymously, and did not object.

### 2.2. [^68^Ga]Ga-PSMA-11 PET/CT Imaging Procedure

Gallium-68 was obtained from a [^68^Ge]Ge/ [^68^Ga]Ga radionuclide generator (GalliaPharm, Eckert & Ziegler Radiopharma GmBH, Berlin, Germany) and used for radiolabelling of PSMA-11 according to the manufacturer’s instructions (IASON GmbH). Patients did not require specific preparation before the injection. Patients received 1–2 MBq/kg of the radiotracer injected in saline via an infusion line.

Images were acquired using a Gemini TF16 (Philips Medical Systems, Cleveland, OH, USA) or a Biograph mCTflow (Siemens Healthcare, Erlangen, Germany) PET/CT. Both PET/CT scanners included time-of-flight technology. Dynamic images were acquired on the pelvis immediately after [^68^Ga]Ga-PSMA-11 injection (10 images of one-minute duration each) and from vertex to mid-thigh 60 to 90 min after injection. On the Gemini TF16 PET/CT, the pelvis was imaged for 3 min, and every other bed position was imaged for 2 min in 3D mode with a 576 mm FOV and a 144 × 144 matrix. Images were reconstructed from 3 iterations and 33 subsets using the OSEM weighted method. Low-dose CT without contrast-enhancement was performed prior to PET acquisition (120 kVp, 80 mA.s, slice thickness 2.5 mm, pitch 0.813, rotation time 0.5 s, FOV 600 mm). On the Biograph mCTflow PET/CT, the scanning speed was set to 0.7 cm/min over the pelvis and 0.9 cm/min for the rest of the acquisition field. Images were taken in 3D mode with a 780 mm FOV and a 200 × 200 matrix. Images were reconstructed from 2 iterations and 21 subsets using the OSEM weighted method. Low-dose CT without contrast-enhancement was performed prior to PET acquisition (CareDose^®^ automatic modulation for keV and mA.s, slice thickness 2 mm, pitch 0.813, rotation time 0.5 s, FOV 500 mm). A harmonization in PET images between the two scanners by using EQ.PET, a NEMA-referenced SUV across technologies was performed [16].

### 2.3. [^68^Ga]Ga-PSMA-11 PET/CT Image Analysis

[^68^Ga]Ga-PSMA-11 PET/CTs were read on-site on the day of image acquisition (routine unmasked reading) by local nuclear physicians with at least 4 years of experience in reading PET/CT and 6 months of experience in reading [^68^Ga]Ga-PSMA-11 PET/CT. An expert nuclear medicine physician with 10 years of experience in reading PET/CT and 4 years of experience in reading [^68^Ga]Ga-PSMA-11 PET/CT, who was blinded to all clinical data, performed a retrospective reading of the [^68^Ga]Ga-PSMA-11 PET/CTs of the patients who met the inclusion criteria (masked retrospective reading). Anonymized images presented in a random order were independently reviewed on a dedicated workstation (Syngo.via, Siemens Healthcare). 

The expert reader assessed uptakes across six anatomical sites and attributed them value on a 3-point qualitative scale according their intensity: 0—no suspicious uptake (at best equal to muscle background); 1—equivocal uptake (between background in muscles and vessels); 2—malignant uptake (higher than background in vessels) [17]. CT images were used for anatomic allocation of a suspicious focus and to facilitate diagnosis. We considered six anatomical sites: prostate/prostatic lodge, pelvic lymph nodes (up to the common iliac lymph nodes), paraaortic lymph nodes, lymph nodes above the diaphragm, bone, and viscera. If at least one suspicious uptake (equivocal or malignant) was detected in an anatomical site, the entire areas was quoted as equivocal or malignant. The intensity of the most intense abnormal uptake was determined by the maximum standardized uptake value (SUVmax) for each anatomical site during the retrospective reading. Based on the results of [^68^Ga]Ga-PSMA-11 PET/CT, we categorized patients as oligometastatic if they present between 1 and 3, 4, or 5 distant malignant uptakes excluding the prostate/prostatic lodge (oligo-3, oligo-4, and oligo-5, respectively); polymetastatic if more than 5 distant malignant uptakes were detected [18].

### 2.4. Follow-Up and Evaluation of [^68^Ga]Ga-PSMA-11 PET/CT Impact on Patient Disease Management 

After imaging, the clinical follow-up was performed for each patient by his referring physician. Clinicians decided for each patient management plan during multidisciplinary meetings dedicated to urological cancers. These multidisciplinary meeting boards were constituted by a urologist, a radiation oncologist, a medical oncologist, a pathologist, a radiologist, and a nuclear medicine physician. They analyzed all clinical data available before and after [^68^Ga]Ga-PSMA-11 PET/CT and then decided the management of the patients. We defined the impact of [^68^Ga]Ga-PSMA-11 PET/CT as any change in management decided by clinicians during the multidisciplinary meeting triggered by [^68^Ga]Ga-PSMA-11 PET/CT. 

The treatment carried out on the patient was considered to be effective if the PSA declined by more than 50% (compared to the baseline value) following treatment modification or if the PSA remained stable (maximum variation of 10% compared to baseline) on at least 2 assays performed at least 3 weeks apart when surveillance was decided [19].

### 2.5. Standard of Truth

Existence of a PCa lesion was established for each patient according to a composite standard of truth (SOT) based on all clinical data that were available during the follow-up period: histological findings, results of other imaging, follow-up imaging and PSA evolution. Histological findings when available were considered as the strongest criteria. When histological confirmation was not available, SOT criteria were as follows:True-positive if at least 3 criteria were met: the imaging was positive for a location; the patient received targeted treatment for imaging findings; the PSA decreased in response to the targeted treatment; the number or the size of the lesions decreased on follow-up imaging;True-negative if at least 3 criteria were met: the imaging was negative for a location; the patient received targeted treatment on another location; the PSA decreased in response to the targeted treatment; no evolution on follow-up imaging;False-positive if at least 3 criteria were met: the imaging was positive for a location; the location was atypical for a PCa metastasis; the patient received targeted treatment for atypical imaging findings leading to an absence of PSA decrease in response to treatment; the patient received targeted treatment on another location leading to a PSA decrease in response to treatment; persistence and stability of the abnormality on follow-up imaging;False-negative if at least 3 criteria were met: the imaging was negative for a location; the patient received targeted treatment on that location leading (as PCa patients presenting with first BCR after prostatectomy and in whom there is no evidence of distant metastatic disease can be offered salvage radiation therapy according to guidelines [20]); PSA decrease in response to treatment; appearance of a typical abnormality in that location on follow-up imaging.

### 2.6. Statistical Analysis

Data were analyzed using the IBM SPSS software. We considered a *p*-value less than 0.05 to be statistically significant. We performed logistic regression to search for a relationship between [^68^Ga]Ga-PSMA-11 PET/CT positivity (patient-based and per anatomical site) and initial ISUP grade group, initial d’Amico group risk, PSA in surgical patients (closest assay to the [^68^Ga]Ga-PSMA-11 PET/CT), PSA doubling time in months, and PSA velocity in ng/mL/year in all patients. Receiver operating characteristic (ROC) analysis was used to determine area under the curve and cut-off values of PSA parameters in relation to [^68^Ga]Ga-PSMA-11 PET/CT positivity. Comparisons of [^68^Ga]Ga-PSMA-11 PET/CT detection rates (at least one suspicious abnormality suggestive of PCa) and impact on patient management to parameters in relation with [^68^Ga]Ga-PSMA-11 PET/CT positivity were performed by chi-squared test, a Fisher’s exact test, or a Student’s *t*-test according to the type of variable. Detection rates and accuracies between the two PET/CT scanners were compared by chi-squared test. Therapy efficacies when guided or not by [^68^Ga]Ga-PSMA-11 PET/CT imaging were compared via Fisher’s exact test. The agreement between retrospective masked and routine unmasked [^68^Ga]Ga-PSMA-11 PET/CT readings, overall and per anatomical site, were assessed using Cohen’s kappa coefficient (0–0.20: very weak; 0.21–0.40: weak; 0.41–0.60: moderate; 0.61–0.80: strong; 0.81–1.0: very strong). 

## 3. Results

### 3.1. BCR Patient Characteristics 

Between June 2016 and November 2018, 294 consecutive patients who met the inclusion criteria were retrospectively included (Table 1). No eligible patient was excluded. One-hundred and ninety-three patients were presenting with first PCa BCR among whom 159 had prostatectomy. 

The mean time from PCa diagnosis to the first BCR was 42 months [95%CI: 37–46], longer for ISUP 1–2 patients (49 months [95%CI: 43–56]) than for ISUP 3–5 patients (33 months [95%CI: 28–39]) (*p* < 0.001; Student *t* test). 

Sixteen patients were considered lost to follow-up (no follow-up data available). The median duration of follow up after [^68^Ga]Ga-PSMA-11 PET/CT for the 278 assessable patients was 17 months [95%CI: 14–19]. A patient-based SOT was feasible in 176 patients (60%) among whom histological confirmation was available in 27 patients. 

### 3.2. [^68^Ga]Ga-PSMA-11 PET/CT Positivity Rates and Performance

On the 294 patients, 140 (48%) where scanned on the Gemini TF16 and 154 (52%) on the Biograph CTflow.

At least one abnormal focus was found in 237 patients (81%) on routine unmasked reading and in 229 patients (78%) on masked retrospective reading. The overall and per anatomical site detection rates, irrespective of PSA, for readings, as well as the SUVmax of detected foci, are presented in Table 2. 

Based on the masked retrospective reading results, we identified a relationship between [^68^Ga]Ga-PSMA-11 PET/CT detection rates and PSA serum level in surgical patients: 37%, 56%, 71%, and 87% for PSA serum levels of 0.20–0.49 ng/mL (*n* = 57), 0.50–0.99 ng/mL (*n* = 45); 1.00–1.99 ng/mL (*n* = 59), and ≥2.00 ng/mL (*n* = 91), respectively, if equivocal findings were considered negative for malignancy; and 59%, 67%, 81%, and 88%, respectively, if equivocal findings were considered positive for malignancy. 

From the ROC analysis, the best cut-off value of PSA to perform a [^68^Ga]Ga-PSMA-11 PET/CT in surgical patients (*n* = 252) was 1 ng/mL (area under curve of 0.59), whether considering equivocal findings as positive or negative for malignancy. 

Overall, the mean PSA serum value of patients with negative [^68^Ga]Ga-PSMA-11 PET/CT was significantly lower than that of the patients with positive examinations, whether considering equivocal findings as positive for malignancy (1.4 vs. 3.8 ng/mL: *p* = 0.03) or negative for malignancy (1.8 vs. 3.9 ng/mL; *p* = 0.03).

A relationship was found between [^68^Ga]Ga-PSMA-11 PET/CT positivity on bone and initial ISUP grade group. However, the rate of bone foci positive was statistically higher for ISUP 3–5 patients than for ISUP 1–2 patients only when considering equivocal findings as negative for malignancy (14% vs. 25%; *p* = 0.03 ).

We did not find any relationship between [^68^Ga]Ga-PSMA-11 PET/CT positivity and the other tested parameters.

Furthermore, we did not identify a difference in [^68^Ga]Ga-PSMA-11 PET/CT positivity between patients presenting with a first BCR and patients presenting with a second or third episode of BCR (*p* = 0.1).

According to [^68^Ga]Ga-PSMA-11 PET/CT results (retrospective masked reading results, equivocal findings considered positive for malignancy), 65 patients (22%) had no detectable disease (60 surgeries), 34 patients (12%) presented an isolated focus in the prostate/prostatic lodge (24 surgeries), 132 patients (45%) were categorized oligo-3 (116 surgeries), 142 (48%) oligo-4 (125 surgeries), 149 (51%) oligo-5 (130 surgeries), and 46 (16%) had more than five distant malignant foci (37 surgeries). 

The dynamic images acquired on the pelvis provided an additional diagnostic information in 6/294 patients (2%), all of whom had surgery, since an abnormal focus in the prostatic lodge was detected on this acquisition but masked by the physiologic urinary uptake in the bladder on the 60 to 90 min after injection PET acquisition.

The overall and per anatomical site diagnostic performances of [^68^Ga]Ga-PSMA-11 PET/CT based on the 176 patients on whom a SOT was feasible are presented in Table 3.

We found no differences in detection rates or in accuracies between the two scanners.

### 3.3. Impact of [^68^Ga]Ga-PSMA-11 PET/CT on BCR Patient Management and Therapy Efficacy Prediction 

The impact of [^68^Ga]Ga-PSMA-11 PET/CT was assessable for the 278 patients for whom follow-up data were available. Patient disease management changed in 189/278 (68%) cases, and [^68^Ga]Ga-PSMA-11 PET/CT impacted this change in 162 cases (86%), 21 being minor changes (Table 4). 

This impact was statistically higher when PSA was greater than 1 ng/mL in surgical patients (97/145 = 67% vs. 43/95 = 45%; *p* < 0.001), but only when PSA was superior to 2 ng/mL in non-surgical patients (21/31 = 68% vs. 1/7 = 14%). Overall, the impact was statistically higher when PSA doubling time was less than one year (113/176 = 65% vs. 46/95 = 48%: *p* = 0.01), and tended to be higher for ISUP 3–5 patients (83/131 = 63% vs. 77/145 = 53%; *p* = 0.08). 

The therapy efficacy was assessable for 257 patients (87%) for whom sufficient follow-up data were available. Treatments with curative intent consisted in 107 radiation therapies focused on abnormalities detected by [^68^Ga]Ga-PSMA-11 PET/CT, 12 salvage lymphadenectomies, two focal irreversible electroporations, one cryosurgery, and one left orchidectomy (isolated CaP metastasis of the testis histologically proven). Eleven patients with negative [^68^Ga]Ga-PSMA-11 PET/CT were treated by radiation therapy of the prostatic lodge in accordance with the guidelines for salvage radiation therapy after prostatectomy [20]. ADT was started for 67 patients among whom three benefited from novel androgen axis drugs (such abiraterone or enzalutamide). Surveillance was finally decided for 75 patients, while it was only indicated in 70 after multidisciplinary meeting, as five patients, in whom PSA presented a long doubling time, refused the offered treatment. 

The treatment carried out on the patient was considered effective according to the defined criteria in 78% (200/257) of patients overall, 89% (138/155) when guided by [^68^Ga]Ga-PSMA-11 PET/CT versus 61% (62/102) when not guided by [^68^Ga]Ga-PSMA-11 PET/CT (*p* < 0.001). It was considered effective in 84% (112/133) when a treatment with curative intent was performed, 94% (60/64) when ADT was started, and 47% (28/60) when surveillance was decided. The treatment carried out on the patient was considered effective in 85%, 86%, and 87% when a treatment with curative intent was performed in oligo-3, oligo-4, and oligo-5 patients, respectively.

### 3.4. Agreement between Routine Unmasked and Retrospective Masked [^68^Ga]Ga-PSMA-11 PET/CT Readings

The agreements between routine unmasked and retrospective masked readings are presented in Table 2. Overall agreement was strong (k = 0.68). The agreement was moderate for the prostate/prostatic lodge (k = 0.54) and viscera (k = 0.56), strong for lymph nodes above the diaphragm (k = 0.73) and bone (k = 0.74), and very strong for pelvic lymph nodes (k = 0.90) and paraaortic lymph nodes (k = 0.84). Both readings were similar in 92% (272/294) of cases. In the 22 cases for which readings were different, the findings on masked readings were considered more accurate according to the follow-up in 6% (17/294: 10 on prostate/prostatic lodge, 2two on bone, two on lymph nodes above the diaphragm, one on pelvic lymph nodes, one on paraaortic lymph nodes, and one on lung) versus 2% (5/294: three on the prostate/prostatic lodge and two on pelvic lymph nodes) for routine reading findings.

## 4. Discussion

### 4.1. [^68^Ga]Ga-PSMA-11 PET/CT Performances

PSMA-11 labelled with gallium-68 a is the most studied ligand for imaging of PCa, especially patients with BCR, for whom detection rates were largely reported, but this study is one of the largest series presenting [^68^Ga]Ga-PSMA-11 PET/CT performances (i.e., sensitivity, specificity, and accuracy), based on a composite SOT. In the present study, [^68^Ga]Ga-PSMA-11 PET/CT demonstrated an overall positivity rate of 78% for restaging PCa patients with BCR, which is consistent with that of 76% reported in a recent meta-analysis [21]. Our per anatomical site positivity rates were also in agreement with an updated version of this meta-analysis [13], except for the extrapelvic lymph nodes positivity rate as we chose to analyze this location in two separate areas (paraarotic lymph nodes and lymph nodes above the diaphragm). We decided to do this because patient management may significantly differ for PCa recurrence between these locations. We also confirmed the relationship between [^68^Ga]Ga-PSMA-11 PET/CT positivity rate and PSA serum levels that has already been reported several times [22,23,24], observing positivity rates in surgical patients comparable to those reported by Perera et al. [13]. 

In our study, we established a composite SOT for PCa on 176 patients overall and at least a hundred patients for each anatomical area. As a histological confirmation of the detected abnormalities was only available in 27 patients, the SOT was primarily based on clinical follow-up including the PSA response to targeted treatment for imaging findings during a median follow-up period of 17 months. Using those criteria, we found overall sensitivity and specificity of both 70%, lower than that of 86% reported by Perera et al. [21], which were only based on histopathologic correlation with ^68^Ga-PSMA PET/CT abnormal findings and, therefore, did not take into account anatomical areas without abnormal PSMA uptake. [^68^Ga]Ga-PSMA-11 PET/CT accuracies for anatomical areas were above 90%, except for the prostate/prostatic lodge area, for which image reading was impaired by the urinary physiologic uptake in the bladder. To improve reading in this location, we performed dynamic PET acquisition over the pelvis immediately after [^68^Ga]Ga-PSMA-11 injection as it was published that this imaging sequence increases the detection rate of local recurrence [25]. In our study these early images provided additional information in 2% (6/294) of patients, all of whom received surgery, by detecting pathological foci in prostate lodge that were then masked by the urinary physiologic uptake in the bladder during the 60 min post-IV PET images. Considering these results, adding dynamic PET images to [^68^Ga]Ga-PSMA-11 PET/CT imaging protocol should be considered for surgical patients. 

### 4.2. [^68^Ga]Ga-PSMA-11 PET/CT’s Impact on PCa Management and Therapy Efficacy Prediction

We found that [^68^Ga]Ga-PSMA-11 PET/CT impacted patient disease management in 58% of cases, resulting in an increased proportion of treatments with curative intent. These findings are consistent with the 54% reported by a recent meta-analysis [14]. Moreover, we found that [^68^Ga]Ga-PSMA-11 PET/CT’s impact was significantly higher when PSA was greater than 1 ng/mL in surgical patients but only when PSA was greater than 2 ng/mL in non-surgical patients. All patients considered, the impact was significantly higher when PSA doubling time was less than one year and tended to be higher for ISUP 3–5 patients. 

In this study, the treatment carried out on patient was considered effective in 91% of cases when guided by [^68^Ga]Ga-PSMA-11 PET/CT versus 40% when not guided by [^68^Ga]Ga-PSMA-11 PET/CT (*p* < 0.001). We previously reported similar findings in a small cohort of 30 castration-resistant PCa patients restaged by [^68^Ga]Ga-PSMA-11 PET/CT [26]. 

### 4.3. [^68^Ga]Ga-PSMA-11 PET/CT Reading Agreement

In our study we analyzed [^68^Ga]Ga-PSMA-11 PET/CTs according to a three-point scale to introduce diagnostic uncertainty in imaging reading, as pitfalls and equivocal findings are inseparable from any diagnostic procedure. Thus, we found equivocal results in approximately 10% for both routine unmasked and retrospective masked readings, which is relatively low. We also noted that the proportion of equivocal findings decreased when PSA increased. Systematic approaches to the interpretation of PSMA imaging studies, using a five-point scale, were recently proposed to classify imaging findings and better reflects the likelihood of the presence of PCa [27,28]. However, we could not use those approaches for the routine readings that were already performed and chose not to use it for the retrospective readings as we wanted to ensure that evaluation between readings would be comparable. Furthermore, we could not use the 2021 EANM standardized guidelines for PSMA-PET as we conducted this research in 2019–2020.

We found an overall strong agreement between [^68^Ga]Ga-PSMA-11 PET/CT routine unmasked and retrospective masked readings (k = 0.68), which is comparable to a previously report on a more heterogeneous series of 50 patients (k = 0.62) [29]. 

Agreement was moderate (k = 0.54) for the prostate/prostatic lodge, slightly lower than previously reported (k = 0.62) [29], likely because our series reports a large proportion of surgical patients. The reading in the prostate/prostate bed is impaired by the physiological uptake of the urine in the bladder, especially in operated patients. Dynamic images aim to improve the reading in the prostate/prostatic lodge. The unmasked reader may look less attentively at the dynamic images because he relies on the clinical data, such as mpMRI or PSA serum value or doubling time that may influence its diagnosis (more equivocal findings in the prostate/prostatic lodge). The experienced masked reader relies only on the dynamic images to improve its reading in the prostate bed.

We found that agreement was strong to very strong for all lymph nodes areas (k = 0.73 for lymph nodes above the diaphragm, k = 0.84 for paraaortic lymph nodes, and k = 0.90 for pelvic lymph nodes), similar to previously reported values of k = 0.74, considering all lymph node areas [29]. However, we distinguished invasion of the lymph nodes between pelvic and paraaortic regions and above the diaphragm, as we assumed that therapeutic management for involved lymph nodes differed between these areas. We found a strong agreement in readings for bone (k = 0.74), which is comparable to that previously reported [29]. Finally, we found a moderate agreement in readings for viscera (k = 0.56), likely due to the low incidence of visceral metastases in patients with BCR [30] and to atypical locations (like penile, testis, intramedullary spinal cord) or challenging location for imaging (30% of visceral lesions were peritoneal carcinomatosis) [31,32]. 

We assumed that the disagreements between readings might be explain by the higher number of equivocal foci found by the routine unmasked reading. Indeed, the knowledge of clinical parameters such as PSA serum level or doubling time may influence the reading.

### 4.4. Limitations

This study has several limitations. The primary one, shared by most imaging studies addressing the search for metastatic disease, is the lack of sufficient histological proof for most of the suspected metastases, which were primarily characterized based on follow-up data. Indeed, obtaining a histopathological evidence for asymptomatic and possibly benign lesions, or locations that are negative on imaging, is ethically questionable and hardly feasible in practice. We chose to base our SOT on the variation of PSA, excluding patients with a change in their ADT regimen after [^68^Ga]Ga-PSMA-11 PET/CT and on histological findings, if available. In this work, a SOT was feasible for 60% of patients, with a 17-month median duration of follow-up, which allowed us to calculate overall performances of [^68^Ga]Ga-PSMA-11 PET/CT, as well as performances per anatomical areas. Furthermore, we were able to determine from the ROC analysis that the best cut-off values of PSA level to perform a [^68^Ga]Ga-PSMA-11 PET/CT in operated patients was 1 ng/mL. These findings need to be confirmed by other large studies, as most available data report ^68^Ga-PSMA PET/CT detection rates [30,33,34]. 

The second major limitation of this work was its retrospective design. However, this study is one of the larger homogenous cohorts of BCR patients which has evaluated imaging performance based on a composite SOT. 

In this work, we reported [^68^Ga]Ga-PSMA-11 PET/CT performance according to PSA serum levels which were not evaluated the day of the PET, in the same laboratory, but corresponded to the value of the closet assays to the examinations. Thus, we assume that we may have overestimated detection rates, especially for low PSA levels.

In this work, [^68^Ga]Ga-PSMA-11 PET/CT acquisition time varied from 60 to 90 min after [^68^Ga]Ga-PSMA-11 injection, which may be important as increased lesion detection was reported with delayed imaging times up to four hours [35]. However, acquisition times were within the acceptable range of 50 to 100 min that is recommended by the current guidelines for ^68^Ga-PSMA PET/CT [35]. Thus, we assume that the limited variation in acquisition times in our study did not significantly affect the results. 

Finally, another notable feature is that all patients in our study who underwent [^68^Ga]Ga-PSMA-11 PET/CT had previously show no sign of distant metastases at [^18^F]fluorocholine PET/CT and were referred for this reason to [^68^Ga]Ga-PSMA-11 PET/CT based on the French regulation for compassionate use of pharmaceutical, which is authorized on an individual basis by the National Medicine Agency. Therefore, our study population may not reflect that of other international studies, and results may differ due to this selection method. However, we found comparable positivity rates than that reported on larger series and metanalyses [13,36].

Because of these limitations, the promising performances and impact rate on patient disease management of [^68^Ga]Ga-PSMA-11 PET/CT need to be confirmed in a larger prospective study.

## 5. Conclusions

In conclusion, [^68^Ga]Ga-PSMA-11 PET/CT demonstrated reliable performance in locating recurrence sites of prostate cancer and motivated disease management changes in almost two out of three patients. Those performances and impact rates were better when PSA serum level were above 1 ng/mL.

Comparison between routine unmasked and retrospective masked readings demonstrated that this imaging modality is highly reproducible, especially for the detection of pelvic and paraaortic lymph nodes.

The use of PSMA radioligands with PET/CT should be considered, when available, as a first line imaging modality for biochemical recurrence of prostate cancer.

## Figures and Tables

**Table 1 cancers-13-01594-t001:** Patient characteristics.

Parameter	
n	294
Mean age in years	
At prostate cancer diagnosis (range)	61 (42–83)
The day of [^68^Ga]Ga-PSMA-11 PET/CT (range)	68 (43–88)
Initial group according to d’Amico classification	
Low risk	32 (11%)
Intermediate risk	170 (58%)
High risk	70 (24%)
Unknown	22 (7%)
International Society of Urological Pathologists (ISUP) 2014 grade group	
1	47 (16%)
2	106 (36%)
3	98 (33%)
4	23 (8%)
5	17 (6%)
Unknown	2 (1%)
Initial treatment	
Surgery (prostatectomy ± lymph node dissection)	210 (71.5%)
Surgery + adjuvant radiation therapy	42 (14%)
Definitive radiation therapy ± androgen deprivation therapy	27 (9%)
Brachytherapy	14 (5%)
High Intensity Focused Ultrasound	1 (0.5%)
PSA parameters at [^68^Ga]Ga-PSMA-11 PET/CT (closest assay to the examination)	
Mean delay between PSA assay and [^68^Ga]Ga-PSMA-11 PET/CT in weeks	10.5 [9.7–11.3]
Mean serum level in ng/mL in operated patients (*n* = 252)	2.97 [1.96–3.98]
0.20–0.49	57 (23%)
0.50–0.99	45 (18%)
1–1.99	59 (23%)
Greater than 2	91 (36%)
Mean serum level in ng/mL in non-operated patients (*n* = 42)	4.96 [3.60–6.31]
Mean doubling time in months^*^	12.9 [11.4–14.7]
Under 6	102 (36%)
Between 6 and 12	80 (28%)
Above 12	103 (36%)
Mean velocity in ng/mL/year *	2.95 [2.17–3.74]

* Evaluated on 285 patients; 95% confidence intervals are presented between brackets.

**Table 2 cancers-13-01594-t002:** [^68^Ga]Ga-PSMA-11 PET/CT positivity rates in prostate cancer patients investigated due to a biochemical recurrence (irrespective of total prostate-specific antigen serum values). Results of routine unmasked and retrospective masked readings, both by anatomical site and overall, are presented. Median maximum standard uptake values (SUVmax) per anatomical site are presented with their range brackets. Agreement was evaluated with Cohen’s kappa coefficient κ.

*n* = 294	Malignant	Equivocal	Negative	SUVmax [Range]	κ
Overall					
Routine unmasked	202 (69%)	35 (12%)	57 (19%)		
Retrospective masked	202 (69%)	27 (9%)	65 (22%)	-	0.68
Prostate/prostatic lodge					
Routine unmasked	60 (20%)	18 (6%)	216 (74%)		
Retrospective masked	60 (20%)	8 (3%)	226 (77%)	5.3 [1.7–20.9]	0.54
Pelvic lymph nodes					
Routine unmasked	110 (38%)	6 (2%)	178 (61%)		
Retrospective masked	111 (38%)	5 (2%)	178 (61%)	5.9 [1.7–58.3]	0.90
Paraaortic lymph nodes					
Routine unmasked	47 (16%)	3 (1%)	244 (83%)		
Retrospective masked	47 (16%)	2 (1%)	245 (83%)	5.5 [1.8–71.7]	0.84
Lymph nodes above the diaphragm					
Routine unmasked	17 (6%)	12 (4%)	265 (90%)		
Retrospective masked	25 (9%)	7 (2%)	262 (89%)	3.9 [2–19.6]	0.73
Bone					
Routine unmasked	53 (18%)	14 (5%)	227 (77%)		
Retrospective masked	57 (19%)	26 (9%)	211 (72%)	3.4 [1.1–38.6]	0.74
Viscera					
Routine unmasked	18 (6%)	9 (3%)	267 (91%)		
Retrospective masked	20 * (7%)	12 ** (4%)	262 (89%)	6.2 [2.2–18.6]	0.56

*: 7 carcinomatosis, 7 pleura/lung, 2 testis, 1 liver, 1 penile, 1 intramedullary spinal, 1 rectal. **: 4 carcinomatosis, 3 liver, 3 testis, 1 penile, 1 pancreas.

**Table 3 cancers-13-01594-t003:** [^68^Ga]Ga-PSMA-11 PET/CT performances in prostate cancer patients investigated due to a biochemical recurrence (irrespective of total prostate-specific antigen serum values). Results with equivocal findings considered positive for malignancy and with equivocal results negative for malignancy are both presented. Patient-based and region-based analyses with the number of cases on which the standard of truth was feasible.

	Se	Sp	Acc
Overall (*n* = 176)			
Equivocal positive for malignancy	73%	57%	71%
Equivocal negative for malignancy	70%	70%	70%
Prostate/prostatic lodge (*n* = 121)			
Equivocal positive for malignancy	76%	91%	85%
Equivocal negative for malignancy	69%	94%	87%
Pelvic lymph nodes (*n* = 116)			
Equivocal positive for malignancy	90%	98%	94%
Equivocal negative for malignancy	88%	100%	95%
Paraaortic lymph nodes (*n* = 103)			
Equivocal positive for malignancy	100%	99%	99%
Equivocal negative for malignancy	100%	100%	100%
Lymph nodes above the diaphragm (*n* = 101)			
Equivocal positive for malignancy	78%	97%	95%
Equivocal negative for malignancy	56%	98%	94%
Bone (*n* = 109)			
Equivocal positive for malignancy	88%	92%	91%
Equivocal negative for malignancy	88%	95%	94%
Viscera (*n* = 101)			
Equivocal positive for malignancy	78%	97%	95%
Equivocal negative for malignancy	56%	98%	94%

**Table 4 cancers-13-01594-t004:** Impact on patient management. Management scheduled before and in view of [^68^Ga]Ga-PSMA-11 PET/CT, overall, for surgical patients and for non-operated patients. Changes induced by [^68^Ga]Ga-PSMA-11 PET/CT results are highlighted in bold and underlined.

**Overall Management**		**Scheduled (*n* = 278)**			
		**Undecided** ***n* = 82**	**Treatment with Curative Intent** ***n* = 50**	**ADT** ***n* = 23**	**Surveillance** ***n* = 1**23
Indicated after [^68^Ga]Ga-PSMA-11 PET/CT	Treatment with curative intent *n* = 140	**48** + 3 #	**18** + 19 *	**4**	**45** + 3
	ADT *n* = 68	**15** + 3 ##	**11** + 1	**1** + 17 **	**18** + 2
	Surveillance *n* = 70	13	1	1	**2** + 53 ***
**Surgical Patients Management**		**Scheduled (*n* = 240)**			
		**Undecided** ***n* = 71**	**Treatment with Curative Intent** ***n* = 48**	**ADT** ***n* = 18**	**Surveillance** ***n* = 103**
Indicated after [^68^Ga]Ga-PSMA-11 PET/CT	Treatment with curative intent *n* = 127	**41** + 3	**18** + 19	**3**	**40** + 3
	ADT *n* = 57	**14** + 3	**10** + 1	**1** + 14	**13** + 1
	Surveillance *n* = 56	10	0	0	46
**Non-Operated Patients Management**		**Scheduled (*n* = 38)**			
		**Undecided** ***n* = 11**	**Treatment with Curative Intent** ***n* = 2**	**ADT** ***n* = 5**	**Surveillance** ***n* = 20**
Indicated after [^68^Ga]Ga-PSMA-11 PET/CT	Treatment with curative intent *n* = 13	**7**	0	**1**	**5**
	ADT *n* = 11	**1**	**1**	3	**5** + 1
	Surveillance *n* = 14	3	1	1	**2** + 7

ADT: androgen-deprivation therapy; #: in 3 cases imaging was negative and patients were treated by salvage radiation therapy according to guidelines; ##: in 3 cases imaging was negative and patients were treated by ADT because of a rapid PSA doubling time (less than 3 months); *: in 18 cases, imaging triggered a modification of radiotherapy fields by finding lymph node metastases (*n* = 17) or an isolated bone metastasis (*n* = 1); **: in 1 case, imaging found multiple bone metastasis and triggered a modification of the planned ADT regimen (switch for a second-generation ADT); ***: in 2 cases, imaging triggered a biopsy of an abnormal uptake, for which pathology demonstrated normal prostatic tissue (false positive of the imaging on the prostate) and a solitary fibrous tumor (false positive of imaging on viscera).

## Data Availability

The data that support the findings of this study are available from the corresponding author, M.G., upon reasonable request.

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
