# Peer review of "Restaging the Biochemical Recurrence of Prostate Cancer with [68Ga]Ga-PSMA-11 PET/CT: Diagnostic Performance and Impact on Patient Disease Management"

_cancers, 2021, doi:10.3390/cancers13071594_

Round 1

Reviewer 1 Report

This is a retrospective study evaluating the accuracy and clinical impact of PSMA-11 PET/CT in men with biochemical recurrence of prostate cancer.  Like other recent studies, a standard of truth for evaluating diagnostic performance was based on a composite of clinical information obtained after treatment.  PET/CT readings were performed prospectively unmasked and retrospectively by a masked reader.  Accuracy is evaluated on a per-site basis across 6 different types of anatomical sites. Although standard of truth for evaluating accuracy could be ascertained  in only 60% of patients, the authors should be commended on the number of patients included as well as the relatively long duration of followup for determining the SOT and clinical impact. Notably, there were relatively few patients lost to followup, and follow-up information on impact was available from 278 patients. A major finding of the study was that the PET findings led to substantial changes in management in many patients.  

Overall, the manuscript is well-written and understandable.  Some minor details that could be of interest to the readers are missing and could be add added in a revision. Those are shown below.  In addition, there is one major question/issue I would like to bring to the authors to directly address:

In reading the eligibility crtieria, did all patients undergo previous fluorocholine PET/CT within the 3 months prior to PSMA-11 PET/CT? If so, could it be considered that the impact of the findings on PSMA 11 PET/CT are due to additional findings not present on fluorocholine PET/CT, or is the change in management due to combined findings from both fluorocholine PET/CT and PSMA-11 PET/CT together given that they were performed within only 3 months of each other.  Please clarify how many (if any) of the patients had “negative” results on fluorocholine PET/CT before being referred to undergo PSMA11 PET/CT.

Minor points:

2.1 Please indicate whether patients included were consecutive. Were there any patients eligibile that were not included due to missing data or any other reasons?

2.1 Eligibility criteria: Please clarify meaning of “considered in remission” as all the patients have biochemical recurrence defined by criteria #3, are they not all recurrent and not in remission?

Are patients with a previous history of another malignancy that is currently in remission eligible?

Two different PET/CT scanners were used. Were there any significant differences in the accuracy results between the scanners? Was there any attempt to harmonize the images between the scanners (eg. SUV cross calibration etc.).

According the the methods, CT images were only used for localization. Why was CT not also used to aid in diagnosis? With Ga-PSMA, examining the CT characteristics of bone lesions is often helpful in distinguishing between benign and pathologic lesion.

The kappa between masked and unmasked readers were lower for viscera and prostate as compared to LN and bone.  It would be interesting for authors to further discuss or speculate on why this was the case.  Is it just because the viscera and prostate are more difficult to interpret? Was the clinical information really that helpful for the unblinded reader? As there were only 2 readers, as the weaker concordance in the prostate lodge related to one reader’s superior ability to interpret the dynamic PET images?  Please provide experience level of all readers.

Author Response

Point 1: In reading the eligibility crtieria, did all patients undergo previous fluorocholine PET/CT within the 3 months prior to PSMA-11 PET/CT? If so, could it be considered that the impact of the findings on PSMA 11 PET/CT are due to additional findings not present on fluorocholine PET/CT, or is the change in management due to combined findings from both fluorocholine PET/CT and PSMA-11 PET/CT together given that they were performed within only 3 months of each other.  Please clarify how many (if any) of the patients had “negative” results on fluorocholine PET/CT before being referred to undergo PSMA11 PET/CT.

Response: Unfortunately, we totally misspoke this sentence as it was unclear for the 3 reviewers. These patients had a BCR and yet they were negative at fluorocholine PET imaging; for this reason, they were referred to a PSMA PET, based on the French compassionate use regulation for compassionate use of pharmaceutical, which is authorized on an individual basis by the National Medicine Agency. So, it cannot be considered that the PSMA-11 PET/CT findings are due to additional findings to that of fluorocholine PET/CT. However, we agree that because of this reglementary point, it can exist a patient selection bias that is discussed in the limitations section.

We reformulated this sentence the way it was suggested by reviewer 2 in section 2.1.

Point 2: Please indicate whether patients included were consecutive. Were there any patients eligible that were not included due to missing data or any other reasons?

Response: Patients were consecutively included. All eligible patients were included. We specified in sections 2.1 and 3.1.

Point 3: Eligibility criteria: Please clarify meaning of “considered in remission” as all the patients have biochemical recurrence defined by criteria #3, are they not all recurrent and not in remission?

Response: We meant we excluded patients with persistent PSA after prostatectomy (PSA ≥ 0.1 ng/ml according to EAU guidelines) or radiation therapy (nadir PSA < 2 ng/ml with testosterone recovered if previous androgen deprivation therapy according to ASTRO guidelines). We clarified in section 2.1.

Point 4: Are patients with a previous history of another malignancy that is currently in remission eligible?

Response: Yes. According to exclusion criterion 4, only patients with a known active neoplasm other than PCa were excluded. In practice, no patient in this series had another active neoplasm. We assume this is because another malignancy would have been detected by fluorocholine PET/CT and taken care of in priority as recurrent prostate cancer can be managed by ADT during treatment of another disease. We clarified in section 2.1.

Point 5: Two different PET/CT scanners were used. Were there any significant differences in the accuracy results between the scanners? Was there any attempt to harmonize the images between the scanners (eg. SUV cross calibration etc.).

Response: Indeed, we attempted to harmonize PET images between the scanners by using a NEMA-referenced SUV across technologies. However, we had not thought to look if there were differences in results between the scanners. So, 154 examinations were performed on the Siemens scanner and 140 on the Philips scanner. At a patient level, we found no differences in detection rates (chi², p=0.34) or in accuracies (chi², p=0.63 when equivocal foci considered positive for malignancy and 0.62 when equivocal foci considered negative for malignancy). These data were added to the 2.2, 2.6 and 3.2 sections. We thank the reviewer for this relevant question.

Point 6: According the methods, CT images were only used for localization. Why was CT not also used to aid in diagnosis? With Ga-PSMA, examining the CT characteristics of bone lesions is often helpful in distinguishing between benign and pathologic lesion.

Response: We misspoke this sentence. We meant that no diagnostic analyses of CT images were performed as we are not radiologists and as CT acquisitions are not configured for diagnostic purposes. We obviously used CT images to facilitate diagnosis. We reformulated this sentence in section 2.3.

Point 7: The kappa between masked and unmasked readers were lower for viscera and prostate as compared to LN and bone.  It would be interesting for authors to further discuss or speculate on why this was the case. Is it just because the viscera and prostate are more difficult to interpret? Was the clinical information really that helpful for the unblinded reader? As there were only 2 readers, as the weaker concordance in the prostate lodge related to one reader’s superior ability to interpret the dynamic PET images?  Please provide experience level of all readers.

Response: The unmasked readings were performed by multiple local specialists the day of the PET. All masked readings were performed by the same nuclear physician who has a large expertise in PSMA PET/CT. We provide experience level of readers in section 2.3.

The prostate bed and viscera are more difficult areas to interpret that require a higher experience in reading scans. Indeed, the reading in the prostate bed is impaired by the physiological uptake of the urine in the bladder, especially in operated patients. Dynamic images aim to improve the reading in the prostatic bed. The unmasked reader may look less attentively at the dynamic images because he relies on the other clinical data such as mpMRI or PSA serum value or doubling time that may influence its diagnosis. This may explain why unmasked reader tends to make more equivocal findings in the prostate/prostatic loge (Table 2).   The experienced masked reader relies only on the dynamic images to improve its reading in the prostate bed. We speculate in that way as requested in section 4.3.

Concerning viscera, we already discussed that the moderate agreement may be due to the low incidence of visceral metastases in patients with BCR and to atypical or challenging locations

Reviewer 2 Report

Major revision

This is an article about the experience of a single centre regarding the diagnostic performance and impact on patients’ management of [68Ga]PSMA-11 PET/CT. Despite a huge cohort, the manuscript presents several major drawbacks that should be considered also by the Editor, due to the high impact of the Journal “Cancers”.   

1) In literature there are several papers regarding this scenario regarding an already well-established impact of such a technique (also in a more clear and strength form):

Fanti et al EJNMMI 2020 doi: 10.1007/s00259-020-04934-4

Hofman et al Lancet 2020; 395(10231):1208-16

Ceci et al EJNMMI 2021 doi: 10.1007/s00259-021-05245-y

2) The SOT used by authors is definitively unclear and heterogeneous: according to PSMA results, what kind of targeted therapy was used? In how many patients? No chemotherapy at all?

3) Discussion section (lines 305-308). Do the authors consider a PSA follow-up (and only in few cases histopathology confirmation) superior to Histopathology confirmation?

4) The paper warrants a minor English Language revision (avoid the use of extended periods in the whole manuscript, also in the results part)

5) Abstract is unclear (the word “impact” is redundant)

6) Authors should use the EANM guidelines for proper radiopharmaceuticals nomenclature

7) Reference 1 is too old for 2021, please see updated Globocan 2020 (Siegel et al)

8) The definition of PCa BCR in the introduction is wrong also considering the relative cited paper (please use adequate ref.4 definition)

9) In a PCa imaging article is not-acceptable the missing of MRI efforts discussion that is the gold standard for such a pathology

10) It is not clear the timing between the considered PSA and PSMA PET acquisition; it is not clear the timing between Cho PET and PSMA PET acquisition; it is not clear how many patients were imaged with each scanner (authors should also briefly described the scanners’ technology, i.e. TOF? Digital?);

11) In materials and methods, authors affirmed that “included patients considered in remission” and few lines after “currently presenting BCR”.

Author Response

Point 1: Some parts are rather wordy, and a more concise presentation of the findings would be much appreciated.

Response: We attempted to lighten the manuscript. However, we had a lot of results to report. We hope that the lightened manuscript will suit the reviewer.

Point 2: Line 72: “These patients have previously been diagnosed as negative for metastasis based on FCH PET/CT according to the French regulation for performing PSMA-11 PET/CT for compassionate use, authorized on an individual basis by the National Medicine Agency”. This sentence is not fully clear, what I understand is that these patients had a BCR and yet they were negative at FCH imaging; for this reason, they were referred to a PSMA PET, on the basis of the French compassionate use regulation. Is that correct? If that’s the case, I would reformulate, as follows: “These patients had shown no sign of distant metastases at FCH PET/CT; for this reason, they were referred to PSMA PET/CT, based on the French regulation for compassionate use of pharmaceutical, which is authorized on an individual basis by the National Medicine Agency”.

Response: The reviewer is right. We reformulate as suggested and thank the reviewer.

Point 3: Line 88 “According to French regulations, we did not need the approval of an institutional review board for performing this retrospective analysis of already available data”. Did the patients sign an informed consent for the use of their pseudonymized data for research purposes?

Response: This research consists in pooling and making a retrospective analysis of anonymised already existing data. As it implied no intervention on the patient,

according to Article 3 of the French “Code de la Santé Publique” (R.1121-1), it was performed according to ethics, without necessity to refer to any Ethics Committee. It was only required to inform patients that their data collected for the PSMA-11 PET/CT would be analysed and published anonymously, and that they did not object. We clarified in section 2.1.

Point 4: Line 119-121. Should the scale be rather 0-1-2 or 1-2-3? Now it is 0-1-3.

Response: We modified the scale which is now scored “0-1-2”.

Point 5: Line 121. Was the score attribution qualitative or did you measure the background uptake in muscle and vessels? If so, where did you measure these uptake indices?

Response: The score attribution was qualitative. We specified in section 2.3.

Point 6: Line 139. Were there no clinical oncologists at the interdisciplinary meetings? Who decided on cytotoxic treatments (whenever indicated), was it the urologist?

Response: We misspoke this sentence. Those multidisciplinary meetings are dedicated to urological cancers. There is always a clinical oncologist for the multidisciplinary meetings dedicated to cancers, this is mandatory according to the French regulation called “plan cancer”. We corrected in section 2.4.

Point 7: Line 140. Please define “PCa imaging specialist” was it always a nuclear medicine physician or a radiologist? or did these two specialists alternate?

Response: Same formulation error than in the previous point. There are always a radiologist and a nuclear medicine physician at these multidisciplinary meetings dedicated to urological cancers. We corrected in section 2.4.

Point 8: Line 144. Here, it appears that the criteria for adequacy are beyond the scope of your work. In fact, you state that an “adequate” treatment must have produced a biochemical response. However, this endpoint does no longer count as a proxy of “management change” but rather as one of “therapy efficacy prediction”. In fact, the choice could have been correct, based on imaging (e.g., abiraterone for metastatic disease not amenable to surgery) yet the disease could have been refractory to this specific treatment. Please explain why you chose such an endpoint and elaborate on this aspect.

Response: We agree with the reviewer point of view. Assessment of “therapy efficacy” is more appropriate than “adequacy” according to our methodology. We corrected in all the manuscript.

Point 9: Line 158: If a location was negative at PSMA PET (and, according to your inclusion criteria, at conventional CT and FCH imaging as well), why and on what basis was it treated with a targeted treatment?

Response: We understand that this point is ambiguous. PCa patients presenting with first BCR after prostatectomy (n=159 in our series) had a negative imaging (n=41) and were treated by radiation therapy of the prostatic lodge (n=11) in accordance with the guidelines for salvage radiation therapy after prostatectomy, that recommend offering salvage radiotherapy to patients with PSA or local recurrence after radical prostatectomy in whom there is no evidence of distant metastatic disease. In 9/11 cases the PSA properly decreased after the treatment. So, we considered that it was a false negative of PSMA-11 PET/CT in the prostatic lodge in those patients. We detailed treatments carried out on all patients in section 3.3.

We performed a composite standard of truth based on all clinical data available during follow-up, which was of a median duration of 17 months, and not exclusively on PSA evolution. However, in view of the comments of the 3 reviewers, it appears that the way we describe the methodology we used for SOT was unclear. It is important to notice that we aimed to define true positive of imaging and also true negative, false negative and false positive. So, we have completely rewritten the section 2.5 (SOT). We hope it will be clearer for the reviewers.

Point 10: Line 190: Please remove “was”.

Response: We kindly notice that this is the verb of the sentence and that we cannot remove it.

Point 11: Line 192 (Table 1) What are the values in brackets in the second column? Range, IQR, CI…?

Response: Values in brackets are the 95% confidence. We clarified in Table 1.

Point 12: Line 256 (Table 4) Some decisions were changed because of reasons other than PSMA PET (e.g., six “undecided” patients were assigned to treatment with curative intent or to ADT). How were these decisions based?

Response: Three patients were treated by ADT because of a rapid PSA doubling time less than 3 months. Three patients presenting with first BCR after surgery. They were treated by radiation therapy of the prostatic lodge in accordance with the guidelines for salvage radiation therapy after prostatectomy, that recommend offering salvage radiotherapy to patients with PSA or local recurrence after radical prostatectomy in whom there is no evidence of distant metastatic disease. We specified by adding footnotes in Table 4.

Point 13: Line 336 Please remove “lower” (decrease lower is a tautology).

Response: We corrected

Reviewer 3 Report

The work by Fourquet and colleagues investigates the impact of PSMA PET on clinical decision-making in a large series. They conclude that this methodic has a relevant impact on the prostate cancer patients’ management, as treatment was adapted in nearly two-thirds of patients. The manuscript is relatively well written; however, some parts are rather wordy, and a more concise presentation of the findings would be much appreciated. Moreover, there are some methodological concerns (detailed below) that should be addressed before publication can be recommended.

Methods

Line 72: “These patients have previously been diagnosed as negative for metastasis based on FCH PET/CT according to the French regulation for performing PSMA-11 PET/CT for compassionate use, authorized on an individual basis by the National Medicine Agency”.

This sentence is not fully clear, what I understand is that these patients had a BCR and yet they were negative at FCH imaging; for this reason, they were referred to a PSMA PET, on the basis of the French compassionate use regulation. Is that correct? If that’s the case, I would reformulate, as follows: “These patients had shown no sign of distant metastases at FCH PET/CT; for this reason, they were referred to PSMA PET/CT, based on the French regulation for compassionate use of pharmaceutical, which is authorized on an individual basis by the National Medicine Agency”.

Line 88 “According to French regulations, we did not need the approval of an institutional review board for performing this retrospective analysis of already available data”. Did the patients sign an informed consent for the use of their pseudonymized data for research purposes?

Line 119-121. Should the scale be rather 0-1-2 or 1-2-3? Now it is 0-1-3.

Line 121. Was the score attribution qualitative or did you measure the background uptake in muscle and vessels? If so, where did you measure these uptake indices?

Line 139. Were there no clinical oncologists at the interdisciplinary meetings? Who decided on cytotoxic treatments (whenever indicated), was it the urologist?

Line 140. Please define “PCa imaging specialist” was it always a nuclear medicine physician or a radiologist? or did these two specialists alternate?

Line 144. Here, it appears that the criteria for adequacy are beyond the scope of your work. In fact, you state that an “adequate” treatment must have produced a biochemical response. However, this endpoint does no longer count as a proxy of “management change” but rather as one of “therapy efficacy prediction”. In fact, the choice could have been correct, based on imaging (e.g., abiraterone for metastatic disease not amenable to surgery) yet the disease could have been refractory to this specific treatment. Please explain why you chose such an endpoint and elaborate on this aspect.

Line 158: If a location was negative at PSMA PET (and, according to your inclusion criteria, at conventional CT and FCH imaging as well), why and on what basis was it treated with a targeted treatment?

Results

Line 190: Please remove “was”.

Line 192 (Table 1) What are the values in brackets in the second column? Range, IQR, CI…?

Line 256 (Table 4) Some decisions were changed because of reasons other than PSMA PET (e.g., six “undecided” patients were assigned to treatment with curative intent or to ADT). How were these decisions based?

Discussion

Line 336 Please remove “lower” (decrease lower is a tautology)

Author Response

Point 1: In literature there are several papers regarding this scenario regarding an already well-established impact of such a technique (also in a clearer and strength form):

- Fanti et al EJNMMI 2020 doi: 10.1007/s00259-020-04934-4 

- Hofman et al Lancet 2020; 395(10231):1208-16 

- Ceci et al EJNMMI 2021 doi: 10.1007/s00259-021-05245-y

Response: It is true that PSMA-PET detecting rates and impact on management in the setting PCa BCR were already explored. Meta-analyses on these topics, which are cited in this manuscript, were also published (Perera et al, Eur Urol 2020; Han et al, Eur Urol 2018). However, this study is one of the largest series with a relatively long duration of follow-up, that presents PSMA-11 PET/CT performances (ie sensitivity, specificity and accuracy) based on a composite SOT and details induced changes in management and clinical follow-up.

Furthermore, we kindly remark that none of the presented references were regarding the scenario of PCa BCR:

- Fanti et al EJNMMI 2020 doi: 10.1007/s00259-020-04934-4 : Reports an expert consensus on PSMA PET/CT response assessment criteria in prostate cancer.

- Hofman et al Lancet 2020; 395(10231):1208-16 : reports PSMA-11 PET/CT performances and impact for initial staging of high-risk PCa patients.

- Ceci et al EJNMMI 2021 doi: 10.1007/s00259-021-05245-y : was recently published (february 2021) and reports EANM reporting guidelines for PSMA-11. We could not use those guidelines to conduct our research as it was conducted in 2019-2020.

Point 2: The SOT used by authors is definitively unclear and heterogeneous: according to PSMA results, what kind of targeted therapy was used?  In how many patients? No chemotherapy at all?

Response: We performed a composite standard of truth based on all clinical data available during follow-up, which was of a median duration of 17 months, and not exclusively on PSA evolution. However, in view of the comments of the 3 reviewers, it appears that the way we describe the methodology we used for SOT was unclear. So, we have completely rewritten the section 2.5 (SOT). It is important to notice that we aimed to define true positive of imaging and also true negative, false negative and false positive. We hope it will be clearer for the reviewers.

Treatments with curative intent consisted in 107 radiation therapies focused on abnormalities detected by [68Ga]Ga-PSMA-11 PET/CT, 12 salvage lymphadenectomies, 2 focal irreversible electroporations, one cryosurgery and one left orchidectomy (isolated CaP metastasis of the testis histologically proven). We detailed treatments carried out on all patients in section 3.3.  No chemotherapy was performed as chemotherapy is rarely offered before ADT to patients presenting with BCR of prostate cancer.

Point 3:  Discussion section (lines 305-308). Do the authors consider a PSA follow-up (and only in few cases histopathology confirmation) superior to Histopathology confirmation?

Response: We misspoke this sentence. Histological confirmation was of course considered as the strongest criteria for SOT. According to our previous response to point 2, we have rewritten the section 2.5 (SOT), and clarified the sentence in the 4.1 section (discussion).

Point 4: The paper warrants a minor English Language revision (avoid the use of extended periods in the whole manuscript, also in the results part)

Response: The English language in the initial manuscript was edited by professionals. However, modifications that were requested may have impaired the English language quality. For an optimal result, we will ask for a new edition of the English language after all modifications will be accepted.

Point 5: Abstract is unclear (the word “impact” is redundant)

Response: We clarified the abstract.

Point 6:  Authors should use the EANM guidelines for proper radiopharmaceuticals nomenclature

Response: We corrected as requested throughout the manuscript.

Point 7: Reference 1 is too old for 2021, please see updated Globocan 2020 (Siegel et al).

Response: We corrected.

Point 8: The definition of PCa BCR in the introduction is wrong also considering the relative cited paper (please use adequate ref.4 definition)

Response: We corrected as requested.

Point 9: In a PCa imaging article is not-acceptable the missing of MRI efforts discussion that is the gold standard for such a pathology.

Response: We agree that mpMRI of the pelvis is recommended for all PCa patients and the gold standard for local staging in the context of initial staging. However, according to the most recent EAU guidelines, it is not recommended for imaging patients with BCR after prostatectomy and only in addition to the PET/CT after radiation therapy in patients fit for salvage therapy (https://uroweb.org/guideline/prostate-cancer/#6). So, as this research is investigating PCa BCR patient and clearly not comparing PSMA PET/CT performances to that of MRI, we don’t think that it is necessary to discuss the place of MRI compared to that of PET/CT in this clinical setting.

Point 10: It is not clear the timing between the considered PSA and PSMA PET acquisition.

Response: As it was a retrospective analysis of data acquired in clinical routine practice, no PSA assay was performed the day of the scan and we considered the most recent value that was available before the PET. So, a PSA value was available for the 294 patients, with a mean time of 10.5 weeks (95%CI: 9.7-11.3) between the PSA assay and the PSMA-11 PET/CT. We added this result in Table 1 and clarified the paragraph where this limitation is discussed in section 4.4.

Point 11: it is not clear the timing between Cho PET and PSMA PET acquisition.

Response: Unfortunately, we misspoke the sentence that was unclear for the 3 reviewers. These patients had a BCR and yet they were negative at fluorocholine PET imaging; for this reason, they were referred to a PSMA PET, based on the French compassionate use regulation for compassionate use of pharmaceutical, which is authorized on an individual basis by the National Medicine Agency. We did not aim to compare fluorocholine and PSMA-11 PET/CT. The PSMA-11 PET/CT findings cannot be considered as additional findings to that of fluorocholine PET/CT. However, we agree that because of this reglementary point, it can exist a patient selection bias that is discussed in the limitations section. We reformulated this sentence the way it was suggested by reviewer 2 in section 2.1. For the reviewer’s information, the mean delay between fluorocholine and PSMA-11 PET was 3.5 months (95%CI: 3.3-3.8).

Point 12: it is not clear how many patients were imaged with each scanner (authors should also briefly described the scanners’ technology, i.e. TOF? Digital?).

Response: We performed 154 examinations on the Siemens scanner and 140 on the Philips scanner. We attempted to harmonize PET images between the scanners by using a NEMA-referenced SUV across technologies. At a patient level, we found no differences in detection rates (chi², p=0.34) or in accuracies (chi², p=0.63 when equivocal foci considered positive for malignancy and 0.62 when equivocal foci considered negative for malignancy). These data were added to the 2.2, 2.6 and 3.2 sections. We also specified that both scanners integrated TOF technology.

Point 13: In materials and methods, authors affirmed that “included patients considered in remission” and few lines after “currently presenting BCR”.

Response: We meant we excluded patients with persistent PSA after prostatectomy (PSA ≥ 0.1 ng/ml according to EAU guidelines) or radiation therapy (nadir PSA < 2 ng/ml with testosterone recovered if previous androgen deprivation therapy according to ASTRO guidelines). We clarified in section 2.1

Round 2

Reviewer 1 Report

The authors provided acceptable responses to this reviewer's comments.

Reviewer 3 Report

All of my previous comments have been successfully addressed by the authors, thank you.